# Effectiveness of seasonal influenza vaccination in patients with diabetes: protocol for a nested case–control study

Ludovic Casanova,[1,2,3] Sébastien Cortaredona,[1,2] Jean Gaudart,[1] Odile Launay,[4,5] Philippe Vanhems,[4,6] Patrick Villani,[1,2] Pierre Verger[1,2,4]

[1]Aix Marseille Univ, INSERM, IRD, SESSTIM, Sciences Economiques & Sociales de la Santé & Traitement de l'Information Médicale, Marseille, France., Marseille, France
[2]ORS PACA, Observatoire Régional de la Santé Provence-Alpes-Côte d'Azur, Marseille, France
[3]Department of General Practice, Aix Marseille University, Marseille, France
[4]Inserm, F-CRIN, Innovative Clinical Research Network in Vaccinology (I-ReIVAC), Paris, France
[5]Paris Descartes University, Sorbonne Paris Cité, Assistance Publique Hôpitaux de Paris, Paris, France
[6]Infection Control and Epidemiology Unit, Edouard Herriot Hospital, Hospices Civils de Lyon, Lyon, France

**Correspondence to**
Dr Ludovic Casanova;
ludovic.casanova@inserm.fr

## ABSTRACT

**Introduction** Seasonal influenza vaccination (SIV) is recommended for people with diabetes, but its effectiveness has not been demonstrated. All of the available studies are observational and marred with the healthy vaccine bias, that is, bias resulting from the generally better health behaviours practised by people who choose to be vaccinated against influenza, compared with those who do not. This protocol is intended to study the effectiveness of SIV in people with treated diabetes and simultaneously to control for bias.

**Methods and analyses** This case-control study is nested in a historical cohort and is designed to study vaccine effectiveness (VE) assessed by morbidity, mortality and anti-infective drug use. The cohort will comprise a representative sample of health insurance beneficiaries in France and will cover 10 consecutive epidemic seasons. It will include all patients reimbursed three separate times for drugs to treat diabetes. The first study of VE will use reasons for hospitalisation as the primary end point, and the second with the use of neuraminidase inhibitors and of antibiotics as the end points. A case will be defined as any person in the cohort reaching any end point at a given date. The case patient will be matched with the largest possible number of controls (individuals not reaching the end point by this date) according to the propensity score method with an optimal calliper width. A conditional logistic model will be used to estimate ORs to take into account both the matching and the repetition of measurements. The model will be applied separately during and outside of epidemic periods to estimate the residual confounding.

**Ethics and dissemination** The study has been approved by the French Commission on Individual Data Protection and Public Liberties (ref: AT/CPZ/SVT/JB/DP/CR052220). The study's findings will be published in peer-reviewed journals and disseminated at international conferences and through social media.

## INTRODUCTION

Seasonal influenza is an important cause of morbidity and mortality.[1] Its incidence is higher in children and young adults, but the morbidity and mortality it causes increase with age and in the presence of some chronic diseases, including diabetes.[2 3] The WHO recommends seasonal influenza

### Strengths and limitations of this study

► This study will generate new evidence to strengthen public policy recommendations for vaccination of people with diabetes, especially among adults of working age.

► This is the first rigorous assessment of the effectiveness of influenza vaccination using the French National Health Insurance System database permitting analysis of about 40 000 patients over a 10-year period.

► This study explicitly accounts for healthy vaccine bias that arises in observational studies where those patients who are more likely to be vaccinated are also more likely to prioritise other areas of their health. This bias may result in overestimation of vaccine effectiveness. In our study, we hypothesise that matching cases and controls by their propensity for other preventive behaviours may reduce this bias.

► The analysis includes a comparison of a non-epidemic control period to assess the presence of residual confounding associated with this healthy vaccine bias.

► As pneumococcal pneumonia is a frequent complication of influenza, and this vaccine is recommended for most patients with diabetes at risk of influenza, this study also considers the pneumococcal vaccination status for the 5 years before an event.

vaccination (SIV) for patients with diabetes[4] because of their high risk of developing severe complications linked to this infection.[2 5] Most studies report that patients with diabetes and healthy patients have equivalent humoral responses to SIV.[6–8] Nonetheless, because influenza antibody titres do not correlate perfectly with clinical protection,[9] vaccine effectiveness (VE) studies are necessary. Recently, two systematic reviews observed a lack of evidence of SIV effectiveness in patients with diabetes and concluded that observational studies are urgently needed to improve the methodological flaws of earlier studies.[10 11]

The measurement of SIV effectiveness in at-risk populations is subject to several methodological difficulties. Randomised clinical trials are not used to measure the effectiveness of recommended vaccines in the general population for ethical reasons; they have long been used and their recommendation indicates their effectiveness.[12] Seven observational studies[13–19] have been performed specifically in patients with diabetes to measure this effectiveness, two[13 15] in patients younger than 65 years. For reasons of feasibility and cost, none of these studies diagnosed influenza directly, by either PCR or serology. Instead they used indirect outcome measures, such as all-cause hospitalisation or all-cause death: this type of unspecific end point provides greater statistical power and increases classification biases. Moreover, five of these studies did not take history of pneumococcal vaccination into account although influenza and pneumococcal pulmonary infection can present very similar clinical pictures, and vaccinations against these two diseases have common determinants of adherence.[19 20] Finally, all seven studies are subject to the healthy vaccine bias[21 22]; this bias results from the fact that people who are vaccinated against influenza are, on the whole, more attentive to their health, visit their physicians more often and probably have better controlled diabetes than unvaccinated individuals. These confounding factors may explain the lower levels of morbidity and mortality observed in vaccinated individuals, independently of the vaccine's protective effect. Accordingly, vaccinated patients have lower rates of morbidity and mortality during non-epidemic periods (that is, seasons during which the influenza virus is not circulating; several authors describe these lower rates as residual confounding).[10 11]

This article presents the protocol of an observational study which has as its principal objective the assessment of SIV effectiveness in patients with treated type 1 and type 2 diabetes while attempting to minimise the preceding methodological limitations. The protocol conducts two nested case–control studies within the very large cohort of beneficiaries of the principal health insurance fund in France. The first study will analyse VE by the reduction in hospitalisations for clinical pictures suggestive of influenza (composite criterion) and for all-cause hospitalisation, and the second by the reduction in outpatient use of neuraminidase-inhibiting antiviral agents (NAIs) and of the antibiotics indicated for treatment of postinfluenza lung diseases.

## METHODS AND ANALYSES
### Database
The study will be performed by analysing reimbursement data from the French National Health Insurance Fund. Data will be anonymously extracted from the fund's permanent sample of beneficiaries ('Echantillon Généraliste des Bénéficiaires' (EGB)) database. The EGB, created in 2003, is a permanent representative sample of persons affiliated to the French health insurance system, a cohort of beneficiaries for whom information on healthcare use is conserved for 20 years. These data can thus be used to assess longitudinal use patterns for research purposes. It is obtained by a 1/97th national random sampling with stratification for age and sex as described in more detail previously.[23] The identification of persons included in the sample is protected by a two-level cryptographic anonymisation process. The EGB is obtained from the reimbursement databases of the major public health insurance funds (salaried workers), which record information on the healthcare use of nearly 85% of the general population in France. On 1 January 2016, the EGB database included 675 906 individuals for whom it contains the following information: age, sex, all reimbursements for medical expenses for outpatient care and their 'chronic disease status' with its diagnostic codes according to the 10th revision of the International Classification of Diseases (ICD-10). 'Chronic disease status' is attributed to people with specific and expensive chronic diseases defined by the French health insurance system and makes them eligible for 100% reimbursement of the costs of their treatment as described in more detail previously.[24] Since January 2006, the EGB has also provided access to all episodes of hospitalisation in public and private hospitals and to their associated discharge diagnoses, coded according to ICD-10.

### Definition of the study cohort
This historical cohort of people treated for diabetes will be followed from 12 October 2006 (date that the 2006–2007 campaign vaccination started) through 9 October 2016 (day before the start of the 2016–2017 campaign vaccination). Patients will be included only if they were in the database for at least 3 years before their follow-up begins. People treated for diabetes will be identified according to the following criteria, previously used in epidemiological studies: they are listed in the databases as having received at least three deliveries of oral diabetes drugs or insulin on three distinct dates in the 12 months before inclusion.[25] For patients to whom oral diabetes drugs were dispensed in the year before the start of follow-up (prevalent diabetes at inclusion), the inclusion date will be the study start date. For new cases of treated diabetes (incident cases of treated diabetes during the study period), the date of inclusion will be the date of the first delivery of diabetes drugs. Patients who change insurance funds or die during follow-up will be censored.

### Years of follow-up
The maximum duration of follow-up will be 10 years. A year of follow-up will begin on the first day of the SIV campaign of the year n and will end the day preceding the start of the campaign for the year n+1. The French Ministry of Defence sets the dates that SIV campaigns begin each year (table 1). The covariables will be assessed at the beginning of each year of follow-up (baseline variables) and considered stable through the end of that year.

**Table 1** Description of the 10 study years: the date the vaccination campaign began, the characteristics of the epidemic and the control periods

| Year | | Campaign start date* | Epidemic period | | | | | Control period | |
|---|---|---|---|---|---|---|---|---|---|
| | | | Start | End | Duration | Peak† | Serotypes‡ | Start | End |
| 1 | 2006/2007 | 12/10/06 | 2007 w03 | 2007 w09 | 7 | 2007 w06 | AH3N2 | 2007 w18 | 2007 w30 |
| 3 | 2008/2009 | 10/10/08 | 2008 w51 | 2009 w08 | 10 | 2009 w04 | AH3N2 | 2009 w18 | 2009 w30 |
| 4 | 2009/2010 | 01/09/09 | 2009 w37 | 2009 w52 | 16 | 2009 w49 | AH1N1 | 2010 w18 | 2010 w30 |
| 5 | 2010/2011 | 24/09/10 | 2010 w51 | 2011 w07 | 9 | 2011 w01 | AH1N1 | 2011 w18 | 2011 w30 |
| 6 | 2011/2012 | 29/09/11 | 2012 w05 | 2012 w12 | 8 | 2012 w08 | AH3N2 | 2012 w18 | 2012 w30 |
| 7 | 2012/2013 | 28/09/12 | 2012 w51 | 2013 w11 | 13 | 2013 w05 | AH1N1/B | 2013 w18 | 2013 w30 |
| 8 | 2013/2014 | 11/10/13 | 2014 w05 | 2014 w10 | 6 | 2014 w07 | AH1N1/AH3N2 | 2014 w18 | 2014 w30 |
| 9 | 2014/2015 | 16/10/14 | 2015 s03 | 2015 s11 | 9 | 2015 s06 | AH3N2 | 2015 w18 | 2015 w30 |
| 10 | 2015/20165 | 15/10/15 | 2016 s04 | 2016 s14 | 11 | 2016 s11 | B | 2016 w18 | 2016 w30 |

*Established by an official communiqué of the National Health Insurance Fund and the Ministry of Health.
†Weeks of maximum incidence.
‡Serotypes: Dominant or codominant virus types and subtypes circulating during the season. This information is based on variable numbers of samples and uses virus detection techniques of different sensitivity. Moreover, several definitions of codominance may have been used, and this information is provided as a rough guide.

### SIV status

During each year of follow-up, SIV status will be defined as positive 8 days after the day the vaccine was dispensed to the patient at the pharmacy (the minimum interval before protective antibodies appear[26–28]). This delivery will be identified by its code in the anatomical, treatment and chemical (ATC) classification for influenza vaccines (ATC Code J07BBxx). We hypothesise that patients will be vaccinated the day they purchase the vaccine at the pharmacy. A sensitivity analysis will test different intervals between the vaccine purchase date and the vaccination date to verify the impact of this hypothesis on the results.

### Cohort-nested case–control study

A case will be defined as any person in the cohort presenting any end point at a given date. For each case, the controls will be defined as all patients not presenting that end point up to the date it occurred for the case. A patient can be chosen as a control numerous times. Any patient who presents an end point during a given year becomes a case and is thus excluded from the control pool until the beginning of the next year. Controls will be matched to cases according to the propensity score method.[29] The propensity score is a one-dimensional summary of all of the matching variables and will furnish an estimate of the probability of being vaccinated as a function of several specific variables. It will be calculated by logistic regression with vaccination as the dependent variable. The model's explanatory variables will be: the date of cohort entry, to control for variations over time (±6 months), age, sex, adherence to secondary prevention examinations for diabetes follow-up, visits to general practitioners (GPs), history of pneumococcal vaccination, comorbidities, diabetes severity, hospitalisation in the previous year and SIV history. In the most common implementation of propensity score matching, pairs of cases and controls are formed when their propensity scores differ by a prespecified maximum amount. To minimise means or risk differences, we will match for the logit of the propensity score, using callipers of width equal to 0.2 of the SD of the logit of the propensity score.[30] The adjusted OR from this conditional logistic regression will be corrected to approach a relative risk (RR) according to the method of Zhang and Yu.[31] The following formula will be used to calculate the VE:

$$VE(\%) = (1 - RR) * 100$$

### First study: end points based on hospitalisation and its causes

The principal end point will be hospitalisation for a clinical picture suggestive of influenza or an indirect complication (table 2). Specifically, it will be the composite criterion of the ICD-10 codes used by Lau *et al* in 2013 to study VE.[15] It is based on the reasons for hospitalisation of patients referred by private-practice physicians to hospital emergency departments for influenza. This outcome is in line with the frequency of clinical complications associated with influenza. For example in England, among 141293 patients with diagnosed influenza, 9.5% presented clinical complications in the 30 days after diagnosis: 1.5% bronchitis, 0.4% lung disease, 5.5% unspecified upper respiratory tract infection.[32] An Italian study[33] reported a higher complication rate (30%) with the complications distributed similarly. Some of the clinical complications included by Lau *et al* (colds, laryngitis and coughs) are too non-specific; we therefore did not include them in the algorithm. The date of the onset of the criterion will be the first day of hospitalisation. The secondary end point will be death or hospitalisation from all causes combined, excluding planned admissions.

### Second study: end points based on outpatient prescriptions

The principal end point will be the dispensing by a private pharmacy of an NAI (ATC Code: J05AH, oseltamivir or zanamivir). NAIs are systemic antiviral treatments used to

**Table 2** List of ICD-10-CA codes included in the administrative case definition of influenza-like illness*

| Diagnosis* | ICD-10 code |
| --- | --- |
| Sinusitis | J01 or J32 |
| Upper respiratory tract infection | J06.8 or J06.9 |
| Influenza | J09 J10 or J11 |
| Viral pneumonia | J12 |
| Acute bronchitis or bronchitis NOS or obstructive bronchitis | J20 or J40 or J44.8 |
| Bronchiolitis | J21 |
| Acute lower respiratory tract infection, not otherwise specified | J22 |
| COPD with acute lower respiratory tract infection (includes pneumonia) | J44.0 |
| COPD with acute exacerbation | J441 |
| Pleurisy | R09.1 |
| Pneumonia | J13 or J14 or J15 or J16 |
| Pneumonia, organism unspecified | J18 |

*Developed using pilot data from emergency departments in Edmonton, Alberta. ICD codes were extracted from randomly selected cases comprising 15% of all emergency department visits with a main ambulatory care diagnosis of influenza.
COPD, chronic obstructive pulmonary disease; ICD-10, 10th revision of the International Classification of Diseases; NOS, not otherwise specified.

cure seasonal influenza in the 48 hours after its first symptoms in individuals at risk of complications, including patients with diabetes, but is prescribed on an outpatient basis to only approximately 25% of patients for whom it is indicated.[34] This end point makes it possible to identify a portion of the influenza cases receiving outpatient care.

The secondary end point will be the delivery in a private pharmacy of one of the antibiotic treatment recommended[35] as a first-line or second-line treatment for cases of acute, community-acquired postinfluenza lung diseases: amoxicillin, amoxicillin/clavulanic acid (ATC Code: J01CR, excluding the following compounds with the same code: ticarcillin/clavulanic acid, ampicillin/sulbactam and piperacillin/tazobactam), ceftriaxone (ATC Code: J01DD04), pristinamycin (ATC Code: J01FG01), levofloxacin (ATC Code: S01AE05) and telithromycin (ATC Code: J01FA15). These recommendations did not change during the study period.

### Statistical analyses
To take into account the variable number of controls per case, the descriptive statistics will be weighted by the inverse of the number of controls in those analyses. As recommended for the nested case–control study design,[36] a conditional logistic model will be used to calculate the OR to take into account both the matching of individuals in the analysis and the repetition of measurements. Recommendations for use of propensity score matching suggest adjusting for the propensity score, because

matching plus adjustment reduces the previous imbalance between two cases and controls more effectively than matching alone.[37] Adjusting for the propensity score thus improves the estimate of the intervention effect.

This model will be applied separately during epidemic periods and non-epidemic periods to estimate possible residual confounding due to the healthy vaccine bias. Each year, the epidemic period will be defined as the period during which the incidence of influenza-like illnesses (ILIs) exceeds the epidemic thresholds defined from influenza surveillance data in France (table 1).[38–40] The non-epidemic periods will be those during which there is no recorded circulation of the virus: 15 May to 31 July each year since 1985 (websenti.u707.jussieu.fr).

Moreover, the analyses will be stratified for age: they will be performed separately for patients aged 65 years or older and for those younger than 65 years. SIV is doubly indicated for patients with diabetes aged 65 years or older—for both their age and their chronic condition. Vaccination coverage of people with diabetes younger than 65 years is lower than that of older patients,[20] and clear VE data lack for this younger age group, who account for nearly half the patients with diabetes treated pharmacologically in France.[41]

### Mismatch between SIV and circulating viruses
The intensity of epidemics varies from year to year due to antigenic modifications of influenza virus strains that can result in a poor match between the circulating strains and those contained in the vaccine and thus result in variations in SIV effectiveness. For instance, mismatches for the A(H3N2) virus during the 2012–2013 and 2014–2015 seasons may explain the poor vaccine effectiveness observed among the elderly during those seasons.[42] The use of data from 10 epidemic seasons will allow us to construct a 'step by step' sensitivity analysis, by excluding years in decreasing order of the extent of the mismatch. We hypothesise that the VE should increase if statistical power remains sufficient.

### Justification of the variables included in the propensity score calculation
#### Age and sex
The risks of hospitalisation and influenza-related complications increase with age, while influenza incidence decreases with age. Age will be defined in the following classes: (18-35); (35-45); (45-55); (55-65); (65-75); (75-85); >85. Including age in the calculation of the propensity score enables consideration of the immunosenescence that reduces VE in older subjects, through its impairment of response to new antigens and its reduction in immune memory.[43 44] For the sex, men appear to have better SIV coverage[45] and are vaccinated with greater regularity.[20]

#### Adherence to examinations for secondary prevention in diabetes follow-up
Adherence to SIV depends, in part, on the behaviour and attitudes of both the patient and the GP in the follow-up of chronic diseases.[46] Patients with diabetes who are vaccinated against influenza are also more adherent in the

follow-up of their diabetes than those not vaccinated;[47] this can affect both the control and complications of their diabetes. In people with diabetes, inadequate disease control is associated with 80% of all-cause hospitalisation.[13 16] These points partly explain the healthy vaccine bias when the VE is calculated from hospital admissions. A proxy variable for adherence with secondary prevention of people with diabetes will be constructed according to a previously published method.[48] This adherence score ranges from 0 to 5 and attributes a value of 1 for each of these examinations performed during the past 3 years: haemoglobin A1c, microalbuminuria, low-density lipoprotein cholesterol, ocular fundus, podiatrist consultation. This score has been shown to be partially correlated with the frequency of GP visits, but not collinear with it.[49]

## Frequency of visits to a general practitioner in the past year

The probability of vaccination against influenza increases with the number of visits to general practitioners.[20 50] For patients 65 years or older with diabetes, Rodriguez *et al*[19] found a mean of 14 annual GP consultations among the unvaccinated patients versus 20 among those vaccinated. GP consultations can modify patients' healthcare pathway by reducing the risk of hospitalisation, for example, by earlier care for influenza (treatment by NAIs in the first 48 first hours of an ILI[34]). At the same time, it enables the best possible diabetes control through more regular monitoring. Preceding studies of VE show that it is preferable to include it as a continuous variable.[13 15]

## History of pneumococcal vaccination

Patients with diabetes are at increased risk of invasive pneumococcal infections.[51–53] Adherence to SIV and to pneumococcal vaccination have common determinants,[50] with pneumococcal vaccination coverage is best in patients vaccinated against influenza.[19] Without adjustment for history of this vaccination, SIV effectiveness may be overestimated. The official recommendation of routine pneumococcal vaccination in patients with diabetes dates back only to 2009 in France but such vaccination took place before then. To construct this adjustment variable, we will consider as vaccinated all patients with at least one delivery of the pneumococcal vaccine (ATC Code: J07A L01) in the 5 years before the year under consideration (in accordance with French guidelines: period of persistence of the protective antibody level with certainty[54 55]). Pneumococcal vaccination status will be considered positive the day the vaccine is dispensed at the pharmacy. Like influenza vaccination status, pneumococcal vaccination status will not be a baseline value but may vary over a year of follow-up. Because of this variable's importance for potential confounding, an interaction term will be added to the models. Should the latter be significant, an analysis stratified by pneumococcal vaccination will be performed to estimate SIV effectiveness in patients with pneumococcal vaccination within the past 5 years and in those vaccinated either never or not in the past 5 years.

## Diabetes severity

The severity of diabetes is simultaneously positively associated with the risk of hospitalisation for infectious causes,[56] including seasonal influenza,[2 57] and with vaccination coverage.[20] It is therefore a confounding factor. Because the EGB databases do not include any individual clinical or laboratory variable to measure diabetes severity directly, it will be approached indirectly by two different variables:

► The intensity of the treatments prescribed: a single oral diabetes medication for mild diabetes, two different such medications for moderate diabetes and more than two different drugs and/or insulin for severe diabetes;

► Time since diagnosis of diabetes is associated with the risks of influenza complications and of immunodeficiency secondary to chronic hyperglycaemia. A proxy variable for this duration will be the estimated as of 2003 (date of earliest EGB data).

## Comorbidities

Patients with diabetes and at least one comorbidity have both better vaccination coverage[20] and a higher risk of influenza complications: if this 'confounding by indication' is not taken into account, it can result in underestimating SIV effectiveness.[22] For each patient, we have diagnoses (coded with ICD-10) associated with potential chronic disease status in addition to diabetes. To use this variable most productively, we plan to regroup them into four major disease groups (box 1): respiratory diseases, cardiovascular diseases, kidney diseases and neoplasms and/or immunodeficiencies.

## Hospitalisation in the preceding year

Adjustment for this variable is also intended to improve control for confounding by indication. Hospitalisation during the preceding year can result in behavioural changes in subsequent years.[58] We can expect an excess of hospitalisations in vaccinated patients, associated not with influenza complications but rather with fragility related to health status. This variable will be constructed dichotomously and separately for each year of follow-up: no hospitalisation or at least one hospitalisation for any cause (excluding planned hospitalisations) during the year preceding the year of follow-up considered.

## History of SIV

Currently, the literature about the effect of repeated SIV is contradictory. A meta-analysis published in 1999 suggested that repeated SIV does not affect its effectiveness.[59] A recent study of vaccinated patients showed a reduction in influenza-positive respiratory infections that was greatest among those not vaccinated for the preceding 5 years.[60] Nonetheless, other work has shown that the serological protection against influenza caused by vaccination may persist from 1 year to another if the virus does not mutate,[57] as it regularly does in practice. Accordingly, one study observed that mortality fell by 15% more in the group regularly vaccinated against influenza

**Box 1    Classification of diseases that can be covered by chronic disease status, in four risk categories, according to the recommendations of the high council for public health (HCSP) with the number of the corresponding condition or disease**

**Respiratory diseases**
► 9 severe forms of neurological or muscular conditions (including myopathy), severe epilepsy
► 14 severe chronic respiratory failure
► 18 cystic fibrosis
► 20 paraplegia
► 25 multiple sclerosis
► 29 active tuberculosis.

**Cardiovascular diseases**
► 1 disabling stroke
► 3 chronic arterial disease with ischaemic events
► 5 severe heart failure, serious arrhythmia, severe heart valve defects, severe congenital heart disease
► 13 coronary disease.

**Kidney failure**
► 19 severe chronic nephropathy and primary nephrotic syndrome.

**Neoplasms and/or immune deficiencies**
► 2 bone marrow failure and other chronic cytopaenia
► 6 chronic active liver disease and cirrhosis
► 7 severe primary immunodeficiency, requiring prolonged treatment, HIV infection
► 21 polyarteritis nodosa, acute disseminated lupus erythematosus, progressive generalised scleroderma
► 22 severe progressive rheumatoid arthritis
► 28 effects of organ transplantation
► 30 malignant tumours, malignant lymphatic or haematopoietic condition.

than in the newly vaccinated group.[61] We have therefore chosen to adjust for the presence of SIV during the 2 years preceding the year under consideration, as earlier studies have done.[13]

## DISCUSSION
### Strengths and limitations of the study design
The design of a case–control study nested in a cohort is a sampling technique based on the principle of sampling within the group at risk ('risk set sampling') and simplifies consideration of exposures and covariables that vary over time. This design optimises the use of the available data by making it possible for a patient to be, at different times, a control and a case and thus facilitates the analysis of infrequent events. The hospitalisation rate for influenza in patients with a chronic condition is around 2% in France.[62] This design also enables the study of rare end points. It should also allow separate analyses of data restricted to the epidemic period and to the non-epidemic season despite the resulting reduction in the number of observations. Another useful point of this design is the improved comparability between cases and

controls made possible by propensity score matching, a method producing the lowest mean quadratic error when large numbers of controls are available.[29 63] Propensity score matching allows the comparison of patients with the same probability of being vaccinated and thus reduces the healthy vaccine bias.

Several points led us to rule out using a control period before the epidemic season. In another study, VE off-season was greater in the postseason rather than the preseason. The authors deduced from their finding that the period after the epidemic season would be a better marker of residual confounding and vaccination bias.[15]

The interest in matching with a propensity score compared with strict matching is to give more weight to some variables that better explain vaccination behaviour. For example, adherence probably predicts vaccination behaviour better than sex, so that giving the two equal weight would decrease the quality and utility of the matching. However, we have already tested[64] in a previous article that none of the variables that we plan to use in the matching score was likely to explain SIV behaviour either by itself or when considered jointly in a multiple regression: the risk of overmatching remains low. We nonetheless plan to build an analytical strategy by progressively including the matching variables in the propensity score to assess the impact of the matching variables.

### Strengths and limitations of end points related to hospitalisation and its causes
The principal outcome selected does not resolve the lack of specificity encountered in the preceding studies. Among the influenza complications included in our algorithm are unspecific bacterial superinfections that are common to other viral infections. To limit the lack of specificity, we nonetheless have not included deterioration or decompensation of chronic diseases, considered in some earlier studies. These hospitalisations for decompensated chronic disease are probably more influenced by patients' health behaviours and thus probably more subject to the healthy vaccine bias. We know that our end point is relatively independent of the healthy vaccine bias and helps to minimise residual confounding. It is therefore probable that we will not observe differences in this end point during the non-epidemic season. Inversely the end points of death and all-cause hospitalisation are more sensitive to this bias. Among patients with diabetes aged 65 years or older, SIV also appears to reduce all-cause mortality by 38% or more.[10] Nonetheless, the excess mortality associated with influenza is estimated at 5% to 10%.[65 66] A substantial portion of the reduction above is therefore linked to unobserved factors. Most studies applying this criterion have found that VE persists during the off-season. This end point has been used as a marker of residual confounding; its failure to appear, contrary to earlier studies, will enable us to shed light on how this design affects it.

## Strengths and limitations of the end points based on outpatient medication use

In patients with diabetes younger than 65 years, the low rate of hospitalisation for ILI makes it difficult to measure VE. The two studies that studied VE with this criterion in this population found no significant effect.[13 15] No study has yet used outpatient NAI use to measure VE, but evidence indicates its relevance. NAIs are a specific treatment for the influenza virus. NAIs can only be purchased on an outpatient basis if prescribed by a doctor. The specificity of this outcome therefore reflects the specificity of doctors' influenza diagnoses of ILI. In epidemic periods, their specificity for this diagnosis is on the order of 60%–70%).[67 68] Models of chronological series of influenza activity in different countries report strong correlations between the official surveillance of sentinel physicians and pharmacy NAI sales,[69–72] especially for patients of working age. The disadvantages of this end point are associated with the multiple other therapeutic indications for NAIs, including: (1) postexposure prophylaxis of seasonal influenza in adult after contact with a clinically diagnosed case in the family and in other exceptional situations; (2) prophylaxis, for both pandemic situations and seasonal influenza during epidemic periods when the circulating viral strains do not match the vaccine. No data describe the proportion of NAIs dispensed for these two indications. The other limitation is related to the lack of specificity of the initial clinical presentation of influenza. Work has shown that primary-care physicians' NAI prescriptions are suboptimal,[73] but no evidence shows that this bias is differential.

Antibiotic prescriptions for influenza complications are the second principal component of outpatient care for influenza accessible in the database. Susceptibility to secondary infection of the lungs and bronchi by bacteria, for example, *Streptococcus pneumoniae* and *Haemophilus influenzae*, seems to result from increased binding of bacteria to the basal membrane of the respiratory epithelium. This may be the result of direct viral damage.[74] Furthermore, the neuraminidase activity of influenzae viruses might thereby decrease the viscosity of the mucus and increase adherence of pneumococci.[75] Finally, the risk of bacterial infections in patients with diabetes in the presence of influenza is aggravated by the immunodeficiency caused by chronic hyperglycaemia.[76] Numerous studies have shown that seasonal influenza epidemics are strongly associated with increased use of antibiotics, especially aminopenicillins and macrolides.[77] Thus, influenza vaccination should result in a reduction in antibiotic use. This variable has already been used as an indirect outcome for assessing SIV effectiveness: in Ontario (Canada): antibiotic prescriptions fell after the introduction of universal free vaccination against influenza.[78 79]

## Strengths and limitations associated with the database

This will be the first study of the effectiveness of seasonal influenza vaccine performed in France from a nationwide healthcare-related administrative database. Nonetheless, these databases do not provide access to laboratory results when laboratory tests are performed, and these tests are not systematically performed. The studies of the VE of SIV with a direct end point measure, such as PCR for influenza from nasopharyngeal swabs, show that indirect outcome measures, such as ILI, can overestimate VE compared with the former (direct outcomes).[80]

Moreover, it is probable that the quality of coding varies between hospitals. Given that the EGB sample is national and random, these coding variations are probably not differential. This type of bias is likely to induce underestimation of VE, which should be made up for in part by our study's statistical power.

**Contributors** LC drafted this study protocol under the direction of PVE and PV. SC contributed to the data management and construction of the statistical model. JG provided his expertise for the design of the nested case–control study. PVA and OL provided their expertise about vaccination and vaccine effectiveness. All the authors participated in the drafting and revision of the manuscript, were major contributors in interpretation of data and in writing the manuscript and approved the final version and agree to be accountable for the contents and integrity of this manuscript.

**Funding** This research received no specific grant from any funding agency in the public, commercial or not for profit sectors.

**Competing interests** None declared.

**Ethics approval** Article 3 of the decree dated 20 June 2005 concerning the implementation of the inter-regional health insurance information system (updated by the decree dated 11 July 2012) foresaw the establishment of a sample of patients affiliated with the French National Health Insurance System, to be called the general beneficiary sample (EGB). Data for this sample may be conserved for 20 years after the day's date. This panel is administered by the CNAM-TS and has been approved by the CNIL (French Commission on Individual Data Protection and Public Liberties, study registered under number AT/CPZ/SVT/JB/DP/CR052220 of 14 June 2005 and number DP/CR071761 of 28 August 2007. French law does not require authorisation of an ethics committee for the use of routinely collected data by authorised research teams (Official Journal of the Republic of French, 2012). The mixed research unit (AMU/Inserm/IRD) that will supervise this study is so authorised.

**Provenance and peer review** Not commissioned; externally peer reviewed.

**Data sharing statement** Data necessary for the study will be available at the end of 2017

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
