## [Reviewer comments · BMJ Open]

ARTICLE DETAILS

TITLE (PROVISIONAL)	Effectiveness of seasonal influenza vaccination in patients with diabetes: protocol for a nested case-control study
AUTHORS	Casanova, Ludovic; Cortaredona, Sebastien; Gaudart, Jean; Launay, Odile; Vanhems, Philippe; Villani, Patrick; Verger, Pierre

VERSION 1 - REVIEW

REVIEWER	Rodrigo Jimenez Garcia Rey Juan Carlos University Spain
REVIEW RETURNED	26-Feb-2017

GENERAL COMMENTS	An interesting well written manuscript focused on an important question.
--

REVIEWER	Lin Yang The Hong Kong Polytechnic University, Hong Kong SAR, China
REVIEW RETURNED	06-Mar-2017

GENERAL COMMENTS	In this protocol, the authors claim that the previous observational studies on influenza vaccine effectiveness had the limitations of nonspecific outcomes and healthy vaccine bias. However, their study design cannot solve any of these problems. 1. Primary outcome variable should be explicitly listed, not just referring to a previous paper. According to that paper, the outcome includes many diseases/symptoms, such as bronchitis, pneumonia, cold, cough, exacerbations of COPD, pharyngitis and sinusitis, but none of these are specific to influenza. Also influenza code J09-J11 is not included. In fact, most of current studies on vaccine effectiveness have adopted a test-negative design, which is regarded as the best design for an observational study on vaccine effectiveness. I understand that lab data might not be available for this cohort, but at least this should be discussed. Although the large cohort has the advantage of large sample size and long follow-up period, this may cause another problem of outcome accuracy (hospital staff have different coding practice). Also it doesn't make sense why antibiotic treatment is used as one of secondary outcomes, as we know antibiotics should not be used for viral infections. If the authors plan to evaluate the effects of secondary bacterial pneumonia, this has been included in the primary outcome. 2. The authors try to address healthy vaccine bias by matching case control pairs with the similar propensity scores, however, the propensity score is calculated for the probability of being vaccinated this season based on the predictors related to vaccination behavior. It is a bit awkward as vaccination is the exposure variable, and the
--

	effect estimate could be minimized by this over matching. 3. The authors discussed vaccine mismatch in the protocol but surprisingly did not mention how this mismatch shall be addressed in their study. Sensitivity analysis of excluding these mismatch years could be considered, or a time-varying model which allows different effect estimates in match and mismatch years can be used. 4. P6, line13, it normally takes two weeks for seroconversion after vaccination, why the authors use 8 days here? 5. P8-9, the paragraphs on justification of propensity score variables are not necessary, as it is well established that these variables are related to vaccination behavior. 6. Inclusion of pneumococcal vaccine history is an advantage of this study, but it is not clear why only the vaccination of prior five years is considered, as the immunity is expected to last for a longer time.
--	--

REVIEWER	Janet McElhaney Health Sciences North Research Institute 41 Ramsey Lake Road Sudbury, ON P3E 5J1 CANADA
REVIEW RETURNED	07-Mar-2017

GENERAL COMMENTS	The title of this manuscript implies that the analysis has been completed when, in fact, this is a proposal for the analysis. I have no issues with the proposed research question and related analysis except to say that it is important to select control periods that occur following the influenza season rather than in advance of the influenza season. However, I am uncertain as to whether the readership of BMJ Open would be interested in the proposed analysis. Once completed, the results of this study will be of significant interest to the medical community.
--

VERSION 1 – AUTHOR RESPONSE

Reviewer: 1

An interesting well written manuscript focused on an important question.

Reviewer: 2

Comment 1:

In this protocol, the authors claim that the previous observational studies on influenza vaccine effectiveness had the limitations of nonspecific outcomes and healthy vaccine bias. However, their study design cannot solve any of these problems.

Response:

We agree with the reviewer that the outcome based on reasons for hospitalization in our study is not specific. The only possible specific endpoint is PCR testing for influenza for every patient with an influenzalike illness (ILI) hospitalized for a complication, and these tests are simply not performed routinely in healthcare facilities in France or elsewhere. Thus, for an observational study the size of ours and based on a health-related administrative database, we could only use reasons for hospitalization as outcome.

Our outcome is based on that of Lau et al. (2013). Compared with other studies assessing the effectiveness of influenza vaccine in people with diabetes, Lau's definition improved the specificity of

this outcome, as we showed in a recently published literature review.[1] As the table below (from our review) makes clear, the definition we are using does not include, in particular, hospitalization for the deterioration or decompensation of chronic diseases, as several other articles do. Nonetheless, it still includes other unspecific diagnoses (see below).

To avoid any ambiguity on this point, we have modified the discussion to underline that our approach does not solve the issue of the non-specificity of the outcome, although it does attempt to improve it. (page 11; line 383-389)

"The principal outcome selected does not resolve the lack of specificity encountered in the preceding studies. Among the influenza complications included in our algorithm are unspecific bacterial superinfections that are common to other viral infections. To limit the lack of specificity, we nonetheless have not included deterioration or decompensation of chronic diseases, considered in some earlier studies. These hospitalizations for decompensated chronic disease are probably more influenced by patients' health behaviours and thus probably more subject to the healthy vaccinee bias."

For the second component of our study (outcome measures based on outpatient drug purchases), we note that neuraminidase inhibitors (NAIs) for outpatients must be prescribed by a physician. The specificity of this outcome therefore reflects the specificity of the physician's influenza diagnosis for patients with an ILI, which is around 60-70% during influenza epidemic periods.[2-4] Specificity of physicians' diagnoses of influenza is especially low for patients aged younger than 18 years, but our study concerns only adults.

On the other hand, the value of this outcome (like that about antibiotic use, see our response below) is that it enables us to cover a larger population and less serious influenza. The effectiveness of influenza vaccine has not been assessed in this type of situation among the general population of people with diabetes.

We have improved the discussion of the specificity of this outcome in the discussion, adding the following sentences (page 11; line 404-407):

"Neuraminidase inhibitors (NAIs) can only be purchased on an outpatient basis if prescribed by a doctor. The specificity of this outcome therefore reflects the specificity of doctors' influenza diagnoses of ILI. In epidemic periods, their specificity for this diagnosis is on the order of 60-70%).[2,4]"

Comment 2:

1. Primary outcome variable should be explicitly listed, not just referring to a previous paper. According to that paper, the outcome includes many diseases/symptoms, such as bronchitis, pneumonia, cold, cough, exacerbations of COPD, pharyngitis and sinusitis, but none of these are specific to influenza. Also influenza code J09-J11 is not included.

Response:

In fact, the list of diseases and symptoms was presented in Table 2 of the first version of the paper.

The revised version of this table eliminates some of them (see our response above).

In the initial version of Table 2 (as well the revised version), we mention the three ICD-10 codes (J09-J11) to capture the diagnoses directly related to influenza, included in the algorithm:

- J09 Influenza due to identified zoonotic or pandemic influenza virus
- J10 influenza due to identified seasonal influenza virus
- J11 Influenza, virus not identified.

-

To justify this outcome more convincingly, we have added epidemiologic evidence about the

frequency of influenza complications in England and Italy. The relevant tables are copied here (see Excerpt 1 below) and in Italy (Excerpt 2 below) for the reviewers' convenience. They are now discussed in the revised methods section to justify our choices: (page 7; line 211-217)

"This outcome is in line with the frequency of clinical complications associated with influenza.[32–34] For example in England, among 141,293 patients with diagnosed influenza, 9.5% presented clinical complications in the 30 days after diagnosis: 1.5% bronchitis, 0.4% lung disease, 5.5% unspecified upper respiratory tract infection.[5] An Italian study [6] reported a higher complication rate (30%,) with the complications distributed similarly. Some of the clinical complications included by Lau et al (colds, laryngitis and coughs) are too non-specific; we therefore did not include them in the algorithm."

Excerpt 1

Frequency of complications in patients with influenza (cases) compared with patients without influenza (controls) in the 30 days after diagnosis (United Kingdom 2000)

Meier CR, Napalkov PN, Wegmüller Y, Jefferson T, Jick H. Population-Based Study on Incidence, Risk Factors, Clinical Complications and Drug Utilisation Associated with Influenza in the United Kingdom. *Eur J Clin Microbiol Infect Dis.* 2000;19(11):834–42.

Excerpt 2

Frequency of complications among patients with influenza (Italy 2001)

Sessa A, Costa B, Bamfi F, Bettoncelli G, D'Ambrosio G. The incidence, natural history and associated outcomes of influenza-like illness and clinical influenza in Italy. *Fam Pract.* 2001;18(6):629–34.

Comment 3:

... In fact, most of current studies on vaccine effectiveness have adopted a test-negative design, which is regarded as the best design for an observational study on vaccine effectiveness. I understand that lab data might not be available for this cohort, but at least this should be discussed...

Response:

We agree with your comment. We have added a comment about the lack of laboratory data in the discussion. (page 12; line 434-438)

"This will be the first study of the effectiveness of seasonal influenza vaccine performed in France from a nationwide healthcare-related administrative database. Nonetheless, these databases do not provide access to laboratory results when laboratory tests are performed, and these tests are not systematically performed. The studies of the vaccine effectiveness (VE) of SIV with a direct endpoint measure, such as PCR for influenza from nasopharyngeal swabs, show that indirect outcome measures, such as ILI, can overestimate VE compared with the former (direct outcomes).[7]"

Comment 4:

...Although the large cohort has the advantage of large sample size and long follow-up period, this may cause another problem of outcome accuracy (hospital staff have different coding practice)...

Response:

We agree with the reviewer that coding practices vary between hospitals. Because our study will cover a national sample, it is likely that these differences in practices will lead to bias, but not to differential bias. A non-differential bias is likely to result in underestimating the vaccine's effectiveness. We have added a sentence to the discussion (page 12, line 439-441) on this point: "Moreover it is probable that the quality of coding varies between hospitals. Given that the EGB sample is national and random, these coding variations are probably not differential. This type of bias is likely to induce underestimation of VE, which should be made up for in part by our study's statistical power."

Comment 5:

...Also it doesn't make sense why antibiotic treatment is used as one of secondary outcomes, as we know antibiotics should not be used for viral infections. If the authors plan to evaluate the effects of secondary bacterial pneumonia, this has been included in the primary outcome.

Response:

We agree with the reviewer that antibiotics are not indicated for typical uncomplicated influenza. The choice of this secondary outcome is motivated by the following points:

First, most influenza-related complications are bacterial infections. Those most commonly encountered after primary influenza infection in adults mainly affect the respiratory system. Susceptibility to secondary infection of the lungs and bronchi by bacteria, e.g., *Streptococcus pneumoniae* and *Haemophilus influenzae*, seems to result from increased binding of bacteria to the basal membrane of the respiratory epithelium. This may be the result of direct viral damage [8] or viral activation of receptors on the epithelial cell that can directly bind bacteria, e.g., the receptor for platelet-activating factor.[9] The influenza virus has also been shown to increase susceptibility by reducing neutrophil function.[10] Neuraminidase also cleaves glycolipids, glycoproteins, and oligosaccharides, thereby decreasing the viscosity of the mucus and exposure receptors on the host epithelial cells.[11] The neuraminidase activity of viruses might thereby contribute to the increased adherence of pneumococci that can be observed during viral infections.

Moreover, the risk of bacterial infections in patients with diabetes in the presence of influenza is aggravated by the immunodeficiency caused by chronic hyperglycaemia.[12]

Accordingly, we think it is useful to assess the extent to which influenza vaccination makes it possible to avoid antibiotic prescriptions in people with diabetes, as previously studied in other populations at risk: some reports show that influenza vaccination of these populations causes a reduction in their antibiotic prescriptions.[13,14]

We have modified the following paragraph of the discussion (page 12; line 418-430) in the manuscript to clarify our explanation of this secondary outcome variable:

"Antibiotic prescriptions for influenza complications are the second principal component of outpatient care for influenza accessible in the database. Susceptibility to secondary infection of the lungs and bronchi by bacteria, e.g., *Streptococcus pneumoniae* and *Haemophilus influenzae*, seems to result from increased binding of bacteria to the basal membrane of the respiratory epithelium. This may be the result of direct viral damage.[15] Furthermore the neuraminidase activity of influenzae viruses might thereby decrease the viscosity of the mucus and increase adherence of pneumococci.[11] Finally, the risk of bacterial infections in patients with diabetes in the presence of influenza is aggravated by the immunodeficiency caused by chronic hyperglycemia.[12] Numerous studies have shown that seasonal influenza epidemics are strongly associated with increased use of antibiotics, especially aminopenicillins and macrolides.[16] Thus, influenza vaccination should result in a reduction in antibiotic use. This variable has already been used as an indirect outcome for assessing SIV effectiveness: in Ontario (Canada): antibiotic prescriptions fell after the introduction of universal free vaccination against influenza.[17]"

Comment 6:

2. The authors try to address healthy vaccine bias by matching case control pairs with the similar propensity scores, however, the propensity score is calculated for the probability of being vaccinated this season based on the predictors related to vaccination behavior. It is a bit awkward as vaccination is the exposure variable, and the effect estimate could be minimized by this over matching.

Response:

Our decision to calculate a propensity score for the probability of being vaccinated was carefully thought out.[18] The reviewer is right to raise the question of overmatching, which concerned us as we determined our matching strategy. The choice of variables for the propensity score was guided by our concern to neutralize the bias associated with vaccination, that is, with the fact that people who are vaccinated have better health behaviours than those who do not; this may explain some of the differences in the incidence of disease episodes in the outcomes of vaccinated and unvaccinated subjects. In the regression analyses we conducted in another study of people with diabetes,[19] we observed that none of the variables that we plan to use in the matching score was likely to explain SIV behaviour either by itself or when considered jointly in a multiple regression: the variance explained by vaccination behaviour remains relatively low. Nonetheless, we still plan to build an analytic strategy by progressively including the matching variables in the propensity score to assess the impact of the matching variables (those most strongly associated with the vaccination bias) on the strength of the associations measured.

We have made the following modification in the article, (page 11; line 374-381):

"The interest in matching with a propensity score compared with strict matching is to give more weight to some variables that better explain vaccination behaviour. For example, adherence probably predicts vaccination behaviour better than sex, so that giving the two equal weight would decrease the quality and utility of the matching. However, we have already verified in a previous article [19] that none of the variables that we plan to use in the matching score was likely to explain SIV behaviour either by itself or when considered jointly in a multiple regression: the risk of overmatching remains low. We nonetheless plan to build an analytic strategy by progressively including the matching variables in the propensity score to assess the impact of the matching variables".

Comment 7 :

3. The authors discussed vaccine mismatch in the protocol but surprisingly did not mention how this mismatch shall be addressed in their study. Sensitivity analysis of excluding these mismatch years could be considered, or a time-varying model which allows different effect estimates in match and mismatch years can be used.

Response:

Thank you for this comment which we have added to our protocol. We have taken care to collect data about the quality of the matching between the serotypes included in the seasonal vaccine and serotypes circulating during the epidemic period concerned. We have expressed it as a percentage.

We have added a paragraph in the methods section (page 8; line 259-266) on this point:

« Mismatch between SIV and circulating viruses

The intensity of epidemics varies from year to year due to antigenic modifications of influenza virus strains that can result in a poor match between the circulating strains and those contained in the vaccine and thus result in variations in SIV effectiveness. For instance, mismatches for the A(H3N2) virus during the 2012-2013 and 2014-2015 seasons may explain the poor vaccine effectiveness observed among the elderly during those seasons.[20] The use of data from 10 epidemic seasons will allow us to construct a "step by step" sensitivity analysis, by excluding years in decreasing order of the extent of the mismatch. We hypothesize that the VE should increase if statistical power remains sufficient."

Comment 8 :

4. P6, line13, it normally takes two weeks for seroconversion after vaccination, why the authors use 8 days here?

Response:

No consensus about the time until seroconversion appears in the literature, but it most often appears to be closer to 5-8 days than 15.[21–23] Nonetheless we had indeed planned in the initial version of the manuscript to perform a sensitivity analysis for this variable. We will therefore test the effect of the interval between vaccination and VE.

Comment 9:

5. P8-9, the paragraphs on justification of propensity score variables are not necessary, as it is well established that these variables are related to vaccination behavior.

Response:

We have simplified these paragraphs (page 8-10).

Comment 10 :

6. Inclusion of pneumococcal vaccine history is an advantage of this study, but it is not clear why only the vaccination of prior five years is considered, as the immunity is expected to last for a longer time.

Response:

We selected this period because the vaccination against pneumococci in populations at risk, including people with diabetes, is recommended in France every 5 years.[24] This guideline is based on the following points. First, levels of antibodies to most pneumococcal vaccine antigens remain elevated for at least 5 years in healthy adults. In some persons, antibody concentrations decrease to prevaccination levels by 10 years.[25,26] However, these quantitative measurements of antibodies do not account for the quality of the antibody being produced or the level of functional immune response. The antibody titre probably diminishes at a constant rate from seroconversion for 10 years. Studies of the persistence of its clinical effectiveness over time are rare.

We have modified the following sentence in the methods section (page 9; line 304-307)

“To construct this adjustment variable, we will consider as vaccinated all patients with at least one delivery of the pneumococcal vaccine (ATC Code: J07A L01) in the 5 years before the year under consideration (In accordance with french guidelines : period of persistence of the protective antibody level with certainty [25,26])

Reviewer: 3

Comment 1:

The title of this manuscript implies that the analysis has been completed when, in fact, this is a proposal for the analysis.

Response:

The title of our manuscript is "Effectiveness of seasonal influenza vaccination in patients with diabetes: protocol for a nested case-control study". This title is based on a protocol –on another subject—recently published in the BMJ Open. We thought that a subtitle starting “protocol for a nested case-control study” was sufficient to inform the reader that this is a protocol, a proposal for an analysis.

Comment 2:

I have no issues with the proposed research question and related analysis except to say that it is important to select control periods that occur following the influenza season rather than in advance of the influenza season.

Response:

We fully agree with the reviewer. We have added a paragraph in the methods section (page 11; line 370-373) on this point:

"Several points led us to rule out using a control period before the epidemic season. In another study, VE off-season was greater in the post- rather than the pre-season. The authors deduced from their finding that the period after the epidemic season would be a better marker of residual confounding and vaccination bias.[27]"

Comment 3:

However, I am uncertain as to whether the readership of BMJ Open would be interested in the proposed analysis. Once completed, the results of this study will be of significant interest to the medical community.

Response:

The BMJ Open regularly publishes observational study protocols. We hope that this protocol will be of interest for the readership of this journal given that randomized clinical trials are not feasible in the field of evaluating vaccine effectiveness (for ethical reasons). It is thus important to share information on protocols and important to have them scrutinized by other scientists. But we must acknowledge that the reviewer's question must be answered by the editor.

REFERENCES

- 1 Casanova L, Gobin N, Villani P, et al. Bias in the measure of the effectiveness of seasonal influenza vaccination among diabetics. *Prim Care Diabetes* Published Online First: 8 June 2016. doi:10.1016/j.pcd.2016.05.005
- 2 Aoki FY, Allen UD, Stiver HG, et al. The use of antiviral drugs for influenza: Guidance for practitioners 2012/2013. *The Canadian Journal of Infectious Diseases & Medical Microbiology* 2012;23:e79.
- 3 Boivin G, Hardy I, Tellier G, et al. Predicting influenza infections during epidemics with use of a clinical case definition. *Clin Infect Dis* 2000;31:1166–9. doi:10.1086/317425
- 4 Yang J-H, Huang P-Y, Shie S-S, et al. Predictive Symptoms and Signs of Laboratory-confirmed Influenza: A Prospective Surveillance Study of Two Metropolitan Areas in Taiwan. *Medicine (Baltimore)* 2015;94:e1952. doi:10.1097/MD.0000000000001952
- 5 Meier CR, Napalkov PN, Wegmüller Y, et al. Population-Based Study on Incidence, Risk Factors, Clinical Complications and Drug Utilisation Associated with Influenza in the United Kingdom. *EJCMID* 2000;19:834–42. doi:10.1007/s100960000376
- 6 Sessa A, Costa B, Bamfi F, et al. The incidence, natural history and associated outcomes of influenza-like illness and clinical influenza in Italy. *Fam Pract* 2001;18:629–34.
- 7 Simonsen L, Taylor RJ, Viboud C, et al. Mortality benefits of influenza vaccination in elderly people: an ongoing controversy. *Lancet Infect Dis* 2007;7:658–66. doi:10.1016/S1473-3099(07)70236-0
- 8 Respiratory Viruses Augment the Adhesion of Bacterial Pathogens to Respiratory Epithelium in a Viral Species- and Cell Type-Dependent Manner. <http://jvi.asm.org/content/80/4/1629> (accessed 12 Apr 2017).
- 9 Cundell DR, Gerard C, Idanpaan-Heikkila I, et al. PAF Receptor Anchors Streptococcus Pneumoniae to Activated Human Endothelial Cells. In: Nigam S, Kunkel G, Prescott SM, eds. *Platelet-Activating Factor and Related Lipid Mediators 2*. Springer US 1996. 89–94. doi:10.1007/978-1-4899-0179-8_16
- 10 McNamee LA, Harmsen AG. Both Influenza-Induced Neutrophil Dysfunction and Neutrophil-Independent Mechanisms Contribute to Increased Susceptibility to a Secondary Streptococcus pneumoniae Infection. *Infect Immun* 2006;74:6707–21. doi:10.1128/IAI.00789-06
- 11 Tong HH, Blue LE, James MA, et al. Evaluation of the Virulence of a Streptococcus pneumoniae

Neuraminidase-Deficient Mutant in Nasopharyngeal Colonization and Development of Otitis Media in the Chinchilla Model. *Infect Immun* 2000;68:921–4. doi:10.1128/IAI.68.2.921-924.2000

12 Delamaire M, Maugendre D, Moreno M, et al. Impaired Leucocyte Functions in Diabetic Patients. *Diabetic Medicine* 1997;14:29–34. doi:10.1002/(SICI)1096-9136(199701)14:1<29::AID-DIA300>3.0.CO;2-V

13 Low D. Reducing antibiotic use in influenza: challenges and rewards. *Clin Microbiol Infect* 2008;14:298–306. doi:10.1111/j.1469-0691.2007.01910.x

14 Kwong JC, Maaten S, Upshur REG, et al. The effect of universal influenza immunization on antibiotic prescriptions: an ecological study. *Clin Infect Dis* 2009;49:750–6. doi:10.1086/605087

15 Avadhanula V, Rodriguez CA, Devincenzo JP, et al. Respiratory viruses augment the adhesion of bacterial pathogens to respiratory epithelium in a viral species- and cell type-dependent manner. *J Virol* 2006;80:1629–36. doi:10.1128/JVI.80.4.1629-1636.2006

16 Sun L, Klein EY, Laxminarayan R. Seasonality and Temporal Correlation between Community Antibiotic Use and Resistance in the United States. *Clin Infect Dis* 2012;55:687–94. doi:10.1093/cid/cis509

17 Delamaire M, Maugendre D, Moreno M, et al. Impaired Leucocyte Functions in Diabetic Patients. *Diabet Med* 1997;14:29–34. doi:10.1002/(SICI)1096-9136(199701)14:1<29::AID-DIA300>3.0.CO;2-V

18 Garrido MM, Kelley AS, Paris J, et al. Methods for Constructing and Assessing Propensity Scores. *Health Serv Res* 2014;49:1701–20. doi:10.1111/1475-6773.12182

19 Verger P, Cortaredona S, Pulcini C, et al. Characteristics of patients and physicians correlated with regular influenza vaccination in patients treated for type 2 diabetes: a follow-up study from 2008 to 2011 in southeastern France. *Clin Microbiol Infect* 2015;21:930.e1-9. doi:10.1016/j.cmi.2015.06.017

20 Centers for Disease Control and Prevention (CDC). Interim adjusted estimates of seasonal influenza vaccine effectiveness - United States, February 2013. *MMWR Morb Mortal Wkly Rep* 2013;62:119–23.

21 Halliley JL, Kyu S, Kobie JJ, et al. Peak frequencies of circulating human influenza-specific antibody secreting cells correlate with serum antibody response after immunization. *Vaccine* 2010;28:3582–7. doi:10.1016/j.vaccine.2010.02.088

22 Cox RJ, Brokstad KA, Zuckerman MA, et al. An early humoral immune response in peripheral blood following parenteral inactivated influenza vaccination. *Vaccine* 1994;12:993–9.

23 Sasaki S, He X-S, Holmes TH, et al. Influence of prior influenza vaccination on antibody and B-cell responses. *PLoS ONE* 2008;3:e2975. doi:10.1371/journal.pone.0002975

24 Nouvelles recommandations 2013 - Vax Info.
<http://www.vaxinfo.be/spip.php?article892&lang=fr> (accessed 13 Apr 2017).

25 Fisman DN, Abrutyn E, Spaude KA, et al. Prior pneumococcal vaccination is associated with reduced death, complications, and length of stay among hospitalized adults with community-acquired pneumonia. *Clin Infect Dis* 2006;42:1093–101. doi:10.1086/501354

26 Musher DM, Manof SB, Liss C, et al. Safety and antibody response, including antibody persistence for 5 years, after primary vaccination or revaccination with pneumococcal polysaccharide vaccine in middle-aged and older adults. *J Infect Dis* 2010;201:516–24. doi:10.1086/649839

27 Lau D, Eurich DT, Majumdar SR, et al. Effectiveness of influenza vaccination in working-age adults with diabetes: a population-based cohort study. *Thorax* 2013;68:658–63. doi:10.1136/thoraxjnl-2012-203109